# Ground-dwelling invertebrate diversity in domestic gardens along a rural-urban gradient: Landscape characteristics are more important than garden characteristics

Brigitte Braschler[1]*, José D. Gilgado[1], Valerie Zwahlen[1], Hans-Peter Rusterholz[1], Sascha Buchholz[2,3], Bruno Baur[1]

**1** Section of Conservation Biology, Department of Environmental Sciences, University of Basel, Basel, Switzerland, **2** Department of Ecology, Technische Universität Berlin, Berlin, Germany, **3** Berlin-Brandenburg Institute of Advanced Biodiversity Research (BBIB), Berlin, Germany

* brigitte.braschler@unibas.ch

**Data Availability Statement:** All relevant data are within the paper and its Supporting Information files.

## Abstract

Urbanisation is increasing worldwide and is regarded a major driver of environmental change altering local species assemblages. Private domestic gardens contribute a significant share of total green area in cities, but their biodiversity has received relatively little attention. Previous studies mainly considered plants, flying invertebrates such as bees and butterflies, and birds. By using a multi-taxa approach focused on less mobile, ground-dwelling invertebrates, we examined the influence of local garden characteristics and landscape characteristics on species richness and abundance of gastropods, spiders, millipedes, woodlice, ants, ground beetles and rove beetles. We assume that most of the species of these groups are able to complete their entire life cycle within a single garden. We conducted field surveys in thirty-five domestic gardens along a rural-urban gradient in Basel, Switzerland. Considered together, the gardens examined harboured an impressive species richness, with a mean share of species of the corresponding groups known for Switzerland of 13.9%, ranging from 4.7% in ground beetles to 23.3% in woodlice. The overall high biodiversity is a result of complementary contributions of gardens harbouring distinct species assemblages. Indeed, at the garden level, species richness of different taxonomical groups were typically not inter-correlated. The exception was ant species richness, which was correlated with those of gastropods and spiders. Generalised linear models revealed that distance to the city centre is an important driver of species richness, abundance and composition of several groups, resulting in an altered species composition in gardens in the centre of the city. Local garden characteristics were important drivers of gastropod and ant species richness, and the abundance of spiders, millipedes and rove beetles. Our study shows that domestic gardens make a valuable contribution to regional biodiversity. Thus, domestic urban gardens constitute an important part of green infrastructure, which should be considered by urban planners.

**Funding:** (J.D. Gilgado) received a grant from the Basler Stiftung für experimentelle Zoologie of CHF 1500.–, which he used to pay student assistants for sorting some of the invertebrate samples. The funder had no effect on the design of the study or the decision to publish.

**Competing interests:** The authors have declared that no competing interests exist.

## Introduction

Urbanisation is increasing globally as more and more people move to cities, with a projected population growth of 2.5 billion people in urban areas by 2050 [1]. As a consequence, urban areas are expanding to the detriment of natural and semi-natural areas. Meanwhile, in already built-up areas urban densification threatens remaining green areas [2, 3]. This trend is diminishing the acknowledged benefits of urban green space for biodiversity and human well-being including experience of nature and improved health of urban residents [4–9]. Existing initiatives seek to mitigate this trend by increasing urban green infrastructure. Examples include the city wall circular greenway in Nanjing [10], the Ring Boulevard in Vienna [11, 12], and the Green Belt Berlin established on the former Berlin Wall [13]. Beside carbon storage and sequestration, these elements of green infrastructure provide ecosystem services such as microclimate regulation, improved air quality, water flow regulation, as well as habitat, food and shelter for plants and animals and thereby increase urban biodiversity [14, 15]. Certain elements of green infrastructure, such as greenways, also contribute to the connectivity of otherwise isolated open habitats or woodlots [16–20].

Besides larger public green spaces such as parks, urban forests and greenways, domestic urban gardens in aggregate constitute a considerable share of the overall urban area. Depending on the city this constitutes a significant part of the overall green space: estimates for the UK varied from 35% for Edinburgh to 47% in Leicester [21], while private patios made up 86% of green area in León, Nicaragua [22] as seen in Goddard et al. [15]. In the light of the dramatic biodiversity crisis, habitat provided by public and private urban green space has an increased importance in supporting populations of animal and plant species [23]. For example, urban green space could play an important role in mitigating insect declines [24–26].

In contrast to larger areas of public green space (e.g. parks), areas with gardens (domestic or community gardens) constitute a heterogeneous small-grained mosaic of diverse habitats influenced by different user management practices and individual preferences [27, 28]. Because of the availability of flowers, several studies on pollinators have been conducted in community gardens, allotment gardens or domestic gardens (e.g. [29–31]). In contrast, the biodiversity of ground-dwelling invertebrates of domestic urban gardens has received little attention in domestic gardens, perhaps due to the dispersed ownership [27] and the assumption that only a reduced biodiversity can be recorded in these heavily managed parcels with many alien plants [32]. However, the few published studies surveying the ground-dwelling invertebrate biodiversity of urban domestic and community gardens, reported considerable numbers of individuals and species in various invertebrate groups if data of multiple gardens were combined (e.g. [33] in London, [34] in Sheffield, and [35] in Toledo, Ohio, USA). For many flying invertebrates (e.g. butterflies, hoverflies and wild bees), a single garden may constitute only a part of their home range because they only provide a part of the resources needed by the species [30]. For these mobile species neighbouring gardens and the further surroundings are essential. In contrast, less mobile small invertebrates may complete their entire life cycle within a garden; e.g. minute land snails [36, 37]. Similarly, millipede communities change with landcover over very small distances and seem little affected by fragmentation, as even communities of habitat specialist millipedes have been shown to persist in small habitat remnants or on small islets [38]. This is prevalent among ground-dwelling invertebrates such as gastropods, spiders, woodlice, millipedes, ants and some groups of beetles with predominantly wingless species. These taxonomical groups play important roles in ecosystems functioning such as decomposing: gastropods, woodlice and millipedes; soil improvement and seed dispersal: ants; predation: spiders, ants, ground beetles and rove beetles [39–45]. However, these groups are often overlooked and not actively promoted or intentionally transported by the garden owners [12].

Biodiversity assessments are frequently based on just one or two conspicuous charismatic groups whose response is assumed to reflect the diversity of other groups [46]. This approach is based on the assumption that species richness of various taxonomical groups are intercorrelated in a habitat; an assumption that is frequently not tested. In contrast, a multi-taxon approach provides both a more comprehensive assessment of the overall biodiversity and an estimate for the phylogenetic diversity and offers the opportunity to disentangle group-specific responses [47]. However, such an approach has rarely been used in urban environments (for exceptions see [48]).

In our study, we aimed at analysing the impacts of urbanisation and garden size on the diversity of seven groups of ground-dwelling invertebrates in the city of Basel and its surroundings (Switzerland). We also aimed to identify how landscape and local garden characteristics drive shifts in species composition. In contrast to some studies that consider different habitat types along the gradient, we focused on domestic gardens and thus considered the same habitat type from the rural surrounding to the city centre. Taken together, a sample of private domestic urban gardens represents a wide range of habitat types, with various degrees of management intensity and a huge range of naturalness (abundance of native plant species, presence of wildlife friendly features such as dead wood or stone piles, extensive management of grassland, bushes and hedges). Thus, a sample of private domestic urban gardens offers niches for numerous species with very different requirements. For example Smith et al. [33] showed that just 11 sites in shared private domestic gardens or similar habitats in parks in London harboured a large share of the overall diversity in the UK for several invertebrate groups. Thus, we predict that (1) our sample of 35 private domestic gardens in combination harbours a considerable share of the species richness in the examined groups of ground-dwelling invertebrates compared to those recorded in Switzerland as a whole.

Along the urbanisation gradient several factors will influence local species richness in gardens. The impact of these factors may increase on approach to the city centre. The heat island effect, input of pollutants and disturbance by light and noise all increased with increasing density of buildings in the city, and they may reduce the habitat suitability for certain species [3, 19, 49]. Furthermore, active dispersal may be reduced by the increasing distance to the source populations and the low permeability of the increasingly sealed urban environment [50]. Therefore, we hypothesise that (2) species richness of ground-dwelling invertebrates decreases and species composition changes with increasing degree of urbanisation, but different taxonomical groups respond differently to urbanisation because they may have differential sensitivity to increased heat, pollution load and other disturbances.

According to the species-area relationship [51], species richness is expected to depend on garden size even in cities [19]. However, this effect may be overlaid by the impact of urbanisation. We thus hypothesise, that (3) ground-dwelling invertebrate species richness increases with increasing garden size. High habitat and structural diversity as well as a diverse native vegetation provide more habitat niches and more varied food resources and in this way increase ground-dwelling invertebrate species richness [34, 52]. We therefore hypothesise that (4) local garden characteristics, such as habitat diversity and structural diversity, increase the richness and alter the composition of ground-dwelling invertebrates, with different garden characteristics affecting different taxonomical groups to a different extent.

## Material and methods

### Garden selection

The study was conducted in the city of Basel, its suburbs and nearby villages in North-western Switzerland (47° 34' N, 7° 36' E). Total annual precipitation averages 842 mm and annual

mean temperature is 10.5˚C in the city (records from 1981 to 2010, www.meteoswiss.admin.ch). Gardens were selected from a pool of 65 candidates offered in response to public calls at a local conference, a municipal newspaper and a newsletter, as well as through personal contacts of the authors. After having visited all gardens in spring 2018, we chose 35 gardens that reflected a rural-urban gradient and represented both a range of garden sizes (definition see below) and different management types (little to intensively managed) spread along the urbanisation gradient. Further criteria for the garden choice were acceptance of the intended sampling methods by the garden owners and guaranteed daytime access to the gardens. All gardens had a grassland area of at least 4 m$^2$, allowing us to set up traps and hay baits (a further criterion for selection), but they differed in the composition of other habitat types (see below).

The study focused on invertebrate species richness, species composition and abundance of sites in a widespread man-made habitat type, namely domestic gardens. We also recorded data on local and landscape-scale environmental characteristics as explanatory variables. However, we did not collect any personal data on garden owners. We do not present any data that could lead to the identification of single gardens or their owners. Therefore, no ethics review was required. Permission to conduct the fieldwork was thus granted by the garden owners themselves. Swiss law does not require fieldwork permits for these groups of invertebrates outside protected areas.

## Garden and landscape characteristics

We assessed 13 local garden characteristics: total garden area, area with vegetation, grassland area, percentage of grassland, area of shrubs and trees, percentage shrub and tree cover, habitat richness, structural diversity, total native plant species richness, native plant species richness in grassland, length of non-permeable garden border, percentage length of non-permeable garden border, index of permeable garden border (Table 1). As landscape characteristics we determined: percentage of sealed area and percentage of green area, both within a radius of 200 m, and distance to the city centre (Table 1).

Total garden area was retrieved from the databases Geoportal BS <map.geo.bs.ch>, Geo-View BL <geoview.bl.ch> and WebGIS Kanton Solothurn <geo.so.ch/map>; all accessed on 28 March 2019. Using a tape measure we determined the area of the following features in each garden: grassland (all types of lawn and meadow combined), tree cover, shrub cover, flower bed, vegetable bed, semi-sealed area, sealed area, and a category of mixed herbaceous vegetation (Table 1). Semi-sealed area included gravel and cobblestone areas, and areas with flagstones. Single flagstones were included here when larger than 0.5 m$^2$. As a surrogate for habitat richness, we recorded the occurrence of each of the following nine land cover types in each garden: grassland (any type), tree, shrub, hedge, flower bed, vegetable bed, compost heap or bin, dead wood (including fire wood, stumps and branches > 3 cm diameters when of a combined length of at least 3 m), and a combined category for other structures (e.g. pile of stones, pond, nest box, bird feeder, insect hotel). We awarded one point to each of the features present resulting in a potential habitat richness ranging from 1–9.

As a measure of structural diversity of a garden, we calculated the Shannon diversity index for the height of all categories of plants. We measured the height of all shrubs and estimated the height of the trees (accuracy: 1 m) using a measuring pole. The height of plants in the grassland area and in flower and vegetable beds was measured along a transect line for each habitat type separately. Transect lines ran along the long axis of the garden features. We considered plants at intervals of 2 m along the transect line. Sampling effort was thus proportional to the area with vegetation. Structural diversity was assessed in all gardens between 24 July and 20 August 2018.

**Table 1. Definitions of local garden and landscape characteristics and transformation of data in the analyses and transformations used in correlation and regression analyses.**

| | Unit | Transformation for correlations and regressions [1] | Description |
|---|---|---|---|
| **Garden size** | | | |
| Total garden area | m² | log | Total garden area excluding buildings |
| Area with vegetation | m² | sqrt when used as continuous variable, otherwise 3 classes: small (< 100 m²), medium (100–400 m²), large (> 400 m²) | Area covered by any type of vegetation, including semi-sealed areas |
| Grassland area | m² | log | Total area covered by any type of grassland (meadow, lawn, etc.) |
| Percentage of grassland | % | arcsine | Percentage of grassland area of the area with vegetation |
| Area of shrubs and trees | m² | sqrt | Total area covered by shrubs and trees (canopy cover) |
| Percentage shrub and tree cover | % | non-parametric analyses | Percentage of area covered by shrubs and trees of the area with vegetation |
| **Garden habitat diversity** | | | |
| Habitat richness | count | non-parametric analyses[2] | Summed occurrence of nine defined habitat features (see main text) |
| Structural diversity | Shannon index | non-parametric analyses[2] | Shannon diversity of height of trees and shrubs, and plants in grassland, flower and vegetable beds |
| **Naturalness** | | | |
| Total native plant species richness | count | $\log^2$ | Number of native plant species in the area with vegetation |
| Native plant species richness in grassland | count | log | Number of native plant species in the grassland area |
| **Isolation of gardens** | | | |
| Length of non-permeable garden border | m | sqrt | Total length of non-permeable garden border (wall height > 30 cm from the outside) including buildings |
| Percentage length of non-permeable garden border | % | not transformed | Percentage of non-permeable garden border length of the total garden border length |
| Index of permeable garden border | % | not transformed | Index combining weighted length of permeable and semi-permeable garden border expressed as percentage of total border length (see main text) |
| **Landscape characteristics** | | | |
| Percentage of sealed area | % | log when used as continuous variable otherwise 3 classes: low (< 40%), medium (40–60.3 m²), large (> 60.3 m²) | Percentage of sealed area in a radius of 200 m around the garden |
| Percentage of green area | % | arcsine | Percentage of green area in a radius of 200 m around the garden |
| Distance to city centre | m | log when used as continuous variable, otherwise 3 classes: short (< 1.5 km), medium (1.5–4 km), long (> 4 km) | Distance from the garden to the town hall of Basel city |

1 Some variables were transformed before being used in Pearson correlation analyses. Log-transformation, square-root-transformation and arcsine-transformations were tried where appropriate. In cases where variables were significantly non-normally distributed even after transformation, we used non-parametric Spearman correlations instead.

2 For local garden characteristics, which were correlated with garden size, we used the residuals of the relationship variable–total garden area when testing for inter-correlations among variables and when including them in GLM models.

We considered the number of native plant species as a surrogate of the naturalness of a garden. We used two measures: native plant species richness found in the grassland and total native plant species richness per garden. We recorded all native species (including woody species) occurring in the grassland by slowly walking in a zigzag line over the grassland area of a garden. Thus, for this variable, sampling effort was proportional to the size of the grassland area in a garden. For total native plant species richness, native plant species richness in grassland was complemented by recording the native plant species in the other habitat types by slowly walking along the transect lines described above for structural diversity. We did not

assess species richness of horticultural plants because the majority of them were characterized by many varieties, meaning that their diversity is representing another taxonomical rank below species.

We considered isolation of gardens by assessing the permeability of their borders to immigration by ground-dwelling invertebrates. For each garden, we measured the total length of non-permeable garden border. We considered a border as non-permeable when invertebrate immigrants were blocked by a building or a wall with a height from the outside > 30 cm. However, migration by ground-dwelling invertebrates may also be reduced by fences or other semi-permeable borders structures. As semi-permeable features we considered walls 10–30 cm in height from the outside and fences with gaps < 3 cm. Thus, permeable borders were defined as total border length minus non-permeable border and semi-permeable border. We calculated an index combining weighted permeable borders and semi-permeable borders by adding the length of the permeable garden border to the halved length of the semi-permeable garden border and expressing it as percentage of the total garden border length (Table 1).

We used three landscape characteristics as surrogates for the degree of urbanisation: percentage sealed area, percentage green area, and distance to city centre. A commonly used measure for degree of urbanisation is the percentage of sealed area (e.g. [3, 53, 54]). We determined the percentages of both sealed and green area within a radius of 200 m around the centre of each garden. We derived land cover data from satellite images (Google Earth, 2009). We then determined the percentage cover of sealed area (built-up area and traffic infrastructure including semi-sealed areas), and green area (urban green space comprising gardens, parks and allotments etc., areas covered by tree canopies, as well as agricultural land and forest cover) using the pixel counting function of Adobe Photoshop Elements (2019). Finally, we measured the distance of each garden to the city centre represented by the town hall of Basel city.

## Invertebrate surveys

For the biodiversity assessment we considered seven groups of ground-dwelling invertebrates. The groups cover a wide range of feeding strategies and included phylogenetically distant taxa: Gastropoda (snails and slugs), Araneae (spiders), Diplopoda (millipedes), Isopoda (woodlice), Formicidae (ants), Carabidae (ground beetles), and Staphylinidae (rove beetles) excluding the subfamily Pselaphinae. We used pitfall traps and hay bait traps to sample all groups. Additional techniques were employed for three groups (Gastropoda, Diplopoda and Formicidae; see below). Sampling was performed between 31 May and 18 October 2018.

Pitfall traps consisted of plastic cups (5.8 cm diameter) partially filled with a saturated salt solution with detergent added to break the surface tension. We chose this non-toxic preservative because children, domestic animals and other small mammals frequently visited gardens. A rain shelter consisting of a 17 cm x 17 cm plastic square 3 cm above ground protected traps also from interference by larger animals. We placed five pitfall traps in the grassland of each garden. Traps were placed randomly. However, if a garden had more than one distinct area with grassland then traps were assigned to each proportionally, but placed in random locations within each area. To account for seasonal differences in invertebrate activity, traps were operated for one week each in early summer, late summer, and autumn. We randomised the order in which gardens were sampled during each period.

We used hay bait traps to target detritivores and their predators. This method has been recommended for millipedes and centipedes [55], but may also work for other ground-dwelling invertebrate groups. A trap consisted of a 25 cm x 25 cm pocket of plastic net with a mesh size of 2 cm filled with wet hay. We placed five hay bait traps in the grassland of each garden in

such a way that the wet hay was in contact with the upper soil layer. Hay bait traps were installed and resampled at the same times as the pitfall traps. They were also distributed in the grassland patches following the same procedure as the pitfall traps. Upon recovery, we transported the baits in individual plastic bags to the laboratory, and placed the hay in Berlese funnels for 10 days. Specimens from both trap types were transferred to 70% ethanol for further species determination.

Using both trapping methods, we obtained a total of 3,099 spider individuals (pitfalls: 2,803 individuals, 90.4% of individuals; hay baits: 296 individuals, 9.6%). Furthermore, we obtained a total of 13,913 woodlice individuals (pitfalls: 7,484 individuals, 53.8%; hay baits: 6,429 individuals, 46.2%). Both trapping methods revealed a total of 49 ground beetle individuals (pitfalls: 43 individuals, 87.8%; hay baits: 6 individuals, 12.2%) and 1279 rove beetle individuals (pitfalls: 175 individuals, 13.1%; hay baits: 1,104 individuals, 86.9%). These taxa were identified to species level using standard identification keys: spiders [56–59]; woodlice [60, 61]; ground beetles [62]; rove beetles [63–69]. Nomenclature followed World Spider Catalog [70] for spiders, Hopkin [61] for woodlice, Müller-Motzfeld [62] for ground beetles and Schülke & Smetana [71] for rove beetles. Captures for gastropods, millipedes and ants are listed below together with records made by other methods employed for these groups.

We applied four methods to assess the species richness and relative abundance of terrestrial gastropods in each garden [72]. First, we visually searched for living snails and slugs and for empty shells on the ground, in the leaf litter, and under dead wood and stones in each garden for a total of 30 min once in early summer. Second, we collected soil samples including dead plant material (up to 2 cm depth, in total a volume of 1 litre per garden) at 4–6 randomly chosen spots in each garden once in early summer. For the extraction of snails, soil samples were sieved (mesh sizes 5 and 2 mm) and later examined using a binocular microscope. The combination of the two methods allows detection of both large-sized taxa that often occur at low density and micro-species that are cryptic and litter-dwelling [73]. Sampling was complemented by the individuals caught in the pitfall and hay traps (see above). The latter methods mainly attracted slugs, which were underrepresented when only the first two methods were applied. Identification of gastropods followed Kerney et al. [74], and the nomenclature of Turner et al. [75] was applied. We determined a total of 3,427 gastropod individuals to the species level (visual search and soil samples: 1,716 individuals, 50.1%; pitfalls: 1,280 individuals, 37.4%; hay baits: 431 individuals, 12.6%).

To examine species richness and abundance of millipedes, we visually searched for millipedes 30 min in each garden and season (in total 90 min per garden). We considered all habitat types but directed special attention to microhabitats preferred by millipedes, such as compost heaps, leaf litter layer, and the underside of pieces of stone and pots. We sampled a total of 6,888 individuals (visual search: 6,052 individuals, 87.9%; pitfalls 70 individuals, 1.0%; hay baits: 766 individuals, 11.1%). Individuals were identified to species level by comparing the external and gonopod morphology with either the original descriptions, or the keys and descriptions present in Blower [76] of the species reported in Switzerland and surrounding countries by Pedroli-Christen [77], and Kime & Enghoff [78, 79]. The nomenclature followed Kime & Enghoff [78, 79]. In a few cases, determination of juveniles or females was only possible at genus or family level (42 individuals, 0.6% of total individuals).

Pitfall traps were the main method to capture ants (9,326 ants; 71.5%), followed by hay baits (3,717 ants; 28.5%), which were very attractive to some ant species (especially *Solenopsis fugax*, *Myrmecina graminicola* and *Tetramorium* cf. *caespitum*). The species list was complemented by an active search of 15 min during each season (total: 45 min per garden). The search prioritised microhabitats and species not sufficiently sampled with the traps, such as mainly arboreal or subterranean species. Only voucher specimens were collected from large aggregations such

as nests or trails. In total 966 ants were collected during the active search. We identified ants to species level. The key of Seifert [80] was used and nomenclature updated according to recent taxonomic revisions following (www.antweb.org). Because of the aggregated distribution of ant workers in these social insects, which are living in colonies, all analyses were performed using presence/absence data (abundance data were not considered in this group).

## Data analyses

Statistical analyses were performed in R (ver. 3.3.3 and ver. 3.6.1, www.r-project.org) and were carried out separately for the different taxonomical groups with the 35 gardens as replicates. We used observed species richness (hereafter species richness) as a surrogate for total species richness (some gardens harboured only one or two individuals of a taxonomical group rendering rarefaction methods inadequate). However, juvenile spiders could only be identified at family level. In some gardens we recorded juveniles from families not represented by adults. In these cases we also calculated supplemented species richness by adding one species for each such family. Juvenile woodlice were not identified. Thus, species richness of woodlice is only based on adults.

We used Pearson's correlation to examine whether species richness of various groups were inter-correlated. Similarly, we tested whether the local garden characteristics and landscape factors assessed were inter-correlated using Pearson's correlation. However, for variables, which were not normally distributed even after transformation, we used Spearman rank correlations instead (Table 1 and S1 Table).

Based on the percentage cover of sealed area in their surroundings, we classified the gardens into areas with low ($< 40\%$), medium (40–60.3%) or high ($> 60.3\%$) degrees of urbanisation. We also assigned gardens into distance classes depending on their distance to the city centre: short ($< 1.5$ km), medium (1.5–4 km), or long ($> 4$ km) (Table 1). Similarly, we assigned gardens to three size classes based on the area with vegetation: small ($< 100$ m$^2$), medium (100–400 m$^2$) and large ($> 400$ m$^2$). For analyses, we considered landscape characteristics and garden size either as factors (first approach) or as continuous variables (second approach) to examine the potential effects on species richness and abundance. In each model we included only one landscape factor, either distance to city centre or percentage sealed area, because these two factors were not independent.

We applied generalised linear models (GLM) with quasi-Poisson distributed errors (previous analyses revealed overdispersion when Poisson error distribution were used) and log-link function to examine potential effects of landscape characteristics, garden size, their interaction, and various local garden characteristics on species richness of different taxonomical groups. In the second approach, the same model was applied but with continuous variables for landscape characteristics and garden size and without their interaction. The two main factors landscape (distance to city centre or percentage sealed area) and garden size were retained in all models, while a step-wise procedure was followed to obtain the minimal adequate models [81]. As further explanatory variables we originally considered all variables listed in Table 1. However, due to collinearity, we omitted several variables, retaining only one from each group of related variables: total native plant species richness, habitat richness, structural diversity and index of permeable border. The first three variables were correlated with garden size. Therefore, we used residuals of the relationships between the variable and total garden area for the GLM models.

As abundance, we considered the total number of individuals captured for each taxonomical group using all collection methods combined. In spiders and woodlice, we calculated abundance for both adult specimens and for all specimens including unidentified juveniles. We did

not consider ant abundance because of the aggregated nature of ant colonies. Analogous to the analyses for species richness, we used GLM models with the same main factors and explanatory variables (quasi-Poisson distributed errors and log-link function; previous analyses revealed overdispersion when using Poisson error distribution). Stepwise reduction of models was done as described above for species richness.

To examine whether local garden characteristics influence the composition of invertebrate communities at the garden level we applied the permutational multivariate analysis of variance (PERMANOVA using the *adonis* function in the *vegan* package, https://cran.r-project.org/web/packages/vegan/index.html; [82]) with matrices based on Sørensen distances. We used constrained analysis of principal coordinates [83] based on community data to assess whether the composition of various invertebrate communities differed among distance-to-the-city-centre classes. We ran ANOVA-like permutations to test for a significant separation of distance classes in a multivariate space. The same approach was used for sealed area classes. We did not consider ground beetles in this analysis because individuals of this group were only recorded in ten gardens. We ran this analysis twice. First, we used data of all species recorded, and second, we used a data set without singletons. Within invertebrate groups, both analyses revealed very similar results (except for rove beetles). We therefore present only the results based on all species (in rove beetles we present both analyses).

We used the Sørensen similarity index to assess the similarities in species composition among all gardens. We calculated the Sørensen-index for all combinations of each two gardens (595 combinations) for each invertebrate group. To examine the potential effect of landscape characteristics on the similarity in species composition, we assigned the 35 gardens into three distance classes according to their distance to the city centre (see above) and calculated the Sørensen-index for all combinations of each two gardens belonging to the same distance class. We used the same procedure for three classes of sealed area.

## Results

### Garden characteristics

The 35 gardens examined ranged in size from 61–1,379 m$^2$ (mean: 479.5 m$^2$; S2 and S3 Tables). On average 86% of the total garden area was covered by vegetation (mean: 412.1 m$^2$; range: 28.8–1,276.9 m$^2$). Grassland was the dominant vegetation type with 37.1% of the vegetated area (mean grassland area: 165.6 m$^2$; range: 4.0–752.3 m$^2$). Habitat richness ranged from 4–9, the maximum possible, with a mean of 7.7, indicating overall rich habitat diversity in the studied gardens. Structural diversity of gardens ranged from 2.6 to 4.4 (Shannon index; mean: 3.7). As proxies of garden naturalness, we assessed total native plant species of entire gardens and the native plant richness of the grassland area. Total native plant species richness ranged from 14 to 128 (mean: 57.2) and native plant species richness in grassland ranged from 8 to 80 (mean: 32.1).

Most gardens had a large proportion of permeable and semi-permeable borders (S2 and S3 Tables), indicated by the index of permeable garden border (mean: 59.8%; range: 4.4–100.0%). Degree of urbanisation expressed as percentage of sealed area (including semi-sealed) within a radius of 200 m around each garden ranged from 32.8% to 87.0% (mean: 52.9%). Distance to the city centre, ranged from 556 m to 9,516 m (mean: 3,307 m). As a proxy for colonisation probability and landscape connectivity, we considered the percentage of green area within a radius of 200 m, which ranged from 6.8% to 67.2% (mean: 45.4%).

Various garden characteristics were positively correlated with total garden area. Larger gardens had a larger area with vegetation (r = 0.98, P < 0.0001; n = 35 in this and following correlations), more area covered by grassland (r = 0.86, P < 0.0001), more area covered by shrubs

and trees (r = 0.71, P < 0.0001), a higher habitat richness ($r_s$ = 0.47, P = 0.0043) and higher structural diversity ($r_s$ = 0.70, P < 0.0001), as well as a higher plant species richness both in the grassland (r = 0.58, P = 0.0002) and overall (r = 0.56, P = 0.0005). However, independent of garden size, the proportion of grassland area and area covered by shrubs and trees remained stable (grassland: r = 0.18, P = 0.30; shrubs and trees: $r_s$ = 0.31, P = 0.0682). Similarly, the permeability of the borders was not correlated with the total area of the corresponding garden (index of permeability: r = 0.16, P = 0.37).

Total garden area and distance to the city centre were not correlated ($r_s$ = 0.06, P = 0.71). This was mainly due to the fact that we selected both large and small gardens at any distance to the city centre for this study. In contrast, gardens were on average larger in less urbanised areas as shown by the positive correlation of total garden area with the percentage of green area in the surroundings (r = 0.46, P = 0.0058), and correspondingly, by the negative correlation with percentage of sealed area (r = -0.35, P = 0.0368).

### Invertebrate species richness and abundance

In the 35 gardens investigated we recorded overall 39 gastropod species, 52 spider species, 22 millipede species, 10 woodlice species, 29 ant species, 26 ground beetle species and 87 rove beetle species (Table 2). The gardens examined harbour an astonishing share of the of species richness of the corresponding groups known for Switzerland (gastropods 19.5% [84], spiders 5.9% (www.cscf.ch; accessed 12 Nov 2019), millipedes 16.7% [77, 85, 86], woodlice 23.3% of non-aquatic isopod species (cscf; communication by Yves Gonseth), ants 20.9% [87], ground beetles 4.7% (www.cscf.ch; accessed 12 Nov 2019), and rove beetles 6.2% [88].

Species richness varied among gardens (Table 2). Depending on species group, the gardens with the highest diversity harboured 2–8 times more species than the gardens with the least

**Table 2. Species richness per garden (n = 35).**

| Taxonomic group | Species richness | | | | | | Relative abundance[6] | |
|---|---|---|---|---|---|---|---|---|
| | Total | Mean ± SD | Range | Mean percentage ± SD | Range of percentage | Chao 1 (Chao 2) | Mean ± SD | Range |
| | | | | | | All gardens | | |
| Gastropods | 39 | 10.5 ± 4.0 | 5–21 | 26.8 ± 10.2 | 12.8–53.9 | 50.7 (61.1) | 97.9 ± 56.2 | 29–267 |
| Spiders | 52 | 9.3 ± 2.8 | 4–18 | 17.2 ± 5.2 | 7.4–33.3 | 58.1 (66.2) | 30.8 ± 14.5 | 11–80 |
| *Spiders suppl.[1]* | *55* | *11.6 ± 3.0* | *6–20* | *20.4 ± 5.3* | *10.5–35.1* | *NA (NA)* | *88.8 ± 45.3* | *27–190* |
| Millipedes | 22 | 5.6 ± 2.3 | 2–13 | 25.3 ± 10.7 | 9.1–59.1 | 22.0 (22.0) | 196.6 ± 177.4 | 16–650 |
| Woodlice[2] | 10 | 4.1 ± 1.7 | 1–8 | 40.6 ± 17.1 | 10.0–80.0 | 10.0 (10.0) | 398.1 ± 811.1 | 1–1884 |
| Ants[3] | 29 | 7.9 ± 2.2 | 4–13 | 27.4 ± 7.5 | 13.8–44.8 | NA (44) | NA | NA |
| Ground beetles | 26 | 0.9 ± 1.3 | 0–5 | 3.3 ± 5.0 | 0.0–19.2 | 204.5 (244.5) | 1.1 ± 1.9 | 0–8 |
| Rove beetles | 87 | 10.7 ± 5.2 | 5–25 | 12.3 ± 5.9 | 5.7–28.7 | 119.7 (138.0) | 36.4 ± 50.8 | 8–275 |
| Total[4] | 265 | 47.5 ± 9.4 | 34–66 | 18.3 ± 3.6 | 13.1–25.4 | 346.0 (384.9) | 1133.3 ± 976.9 | 400–4928 |
| *Total suppl.[5]* | *268* | *51.2 ± 9.6* | *37–73* | *19.0 ± 3.5* | *13.7–27.0* | *NA (NA)* | *NA* | *NA* |

Percentages refer to the share of species of a group found in single gardens in relation to the total number of species recorded in all gardens for the respective taxonomical group. Supplemented species richness is given in italics.

1 Supplemented spider species richness includes added species for families only represented by juvenile spiders within a garden.

2 Based on adult specimens identified to species level. Including juveniles the relative abundance ranged from 1–4217 individuals.

3 Observed species richness of ants based on pitfall traps and hay baits supplemented by active search. As the latter was not quantitative, indices requiring measures of abundance could not be calculated (NA). For an overall estimate of supplemented ant species richness in all gardens we calculated the incidence-based Chao2 estimator.

4 Total species richness is based on identified adult specimens of all groups.

5 Supplemented total species richness includes added species for families only represented by juvenile spiders in a garden.

6 All gardens were sampled with standardised procedures independent of garden size. Relative abundance is therefore a proxy for the variation in density of different taxonomic groups among gardens.

species (Table 2). Interestingly, a particular garden could contain a large share of the overall number of species recorded in one or a few taxonomical groups but a poor share in other species groups. For example, the garden with the most invertebrate species overall also had the most woodlice species of all gardens and was among the most species-rich gardens when considering gastropods, ants or beetles. However, the same garden was only ranked eleventh out of 35 for spiders and came last for millipedes (S4 and S5 Tables). In relation to the species pool of our 35 gardens, a single garden had on average 18.3% of the total number of species recorded in our study (range: 3.3% of all ground beetle species to 40.6% of all woodlice species; Table 2 and S5 Table). Similarly to species richness, relative abundance of the studied taxonomical groups varied among gardens, with some groups relatively poorly represented in several gardens (Table 2 and S6 Table).

The fact that the same garden had higher than average species richness for some taxonomical groups, but lower than average species richness for other groups, is mirrored by the lack of correlations among the species richness of most invertebrate groups, considering gardens as independent replicates (S1 Table). This indicates that single taxonomical groups are poor estimators of overall biodiversity in private domestic gardens.

## Effects of landscape and local garden characteristics on invertebrate species richness

We used two landscape factors, capturing different aspects of urbanisation. Distance to the city centre is related to the degree of isolation from larger semi-natural areas, while percentage of sealed area within 200 m refers to the quality of the matrix surrounding a particular garden. Distance to city centre affected species richness in gastropods, spiders, millipedes, ants and rove beetles and tended to affect that of woodlice (Fig 1; S7 Table). The shape of the relationship varied depending on the invertebrate group. While spider and ant species richness increased with increasing distance from the city centre, the opposite was true for rove beetles (Fig 1; S7 Table). Gastropod and millipede species richness were also lowest at long distance from the city centre but their richness was highest at medium distance from the city centre (Fig 1; S7 Table). Most of these landscape effects disappeared if the percentage of sealed area rather than distance to the city centre is used in the models, the exception being the high richness of rove beetles in gardens with a high percentage of sealed area in the surroundings (S7 Table).

In the models with distance to city centre, area with vegetation, a measure of garden size, had a positive effect on species richness of ants, and tended to influence species richness of spiders (Fig 1; S7 Table; spiders: u-shaped; spiders supplemented: hump-shaped). In the models considering percentage of sealed area, area with vegetation positively affected the species richness of spiders and ants (S7 Table). No interactions between distance to city centre or percentage of sealed area with area with vegetation were found in any group (Fig 1; S7 Table). Garden border permeability (index of permeable garden border), a component of garden isolation, did not affect species richness of any group (Fig 1; S7 Table).

Characteristics reflecting the naturalness and diversity of the gardens (native plant species richness, habitat richness, structural diversity) affected species richness of different groups to a varying degree. Native plant species richness positively affected species richness of gastropods in models that considered distance to the city centre or percentage of sealed area as classes (Fig 1; S7 Table). A similar positive effect of native plant species richness on species richness of ants was found for models with continuous variables for distance to the city centre and percentage sealed area. Both habitat richness and structural diversity only influenced gastropod richness. Habitat richness positively affected gastropod richness in the model with distance to city centre

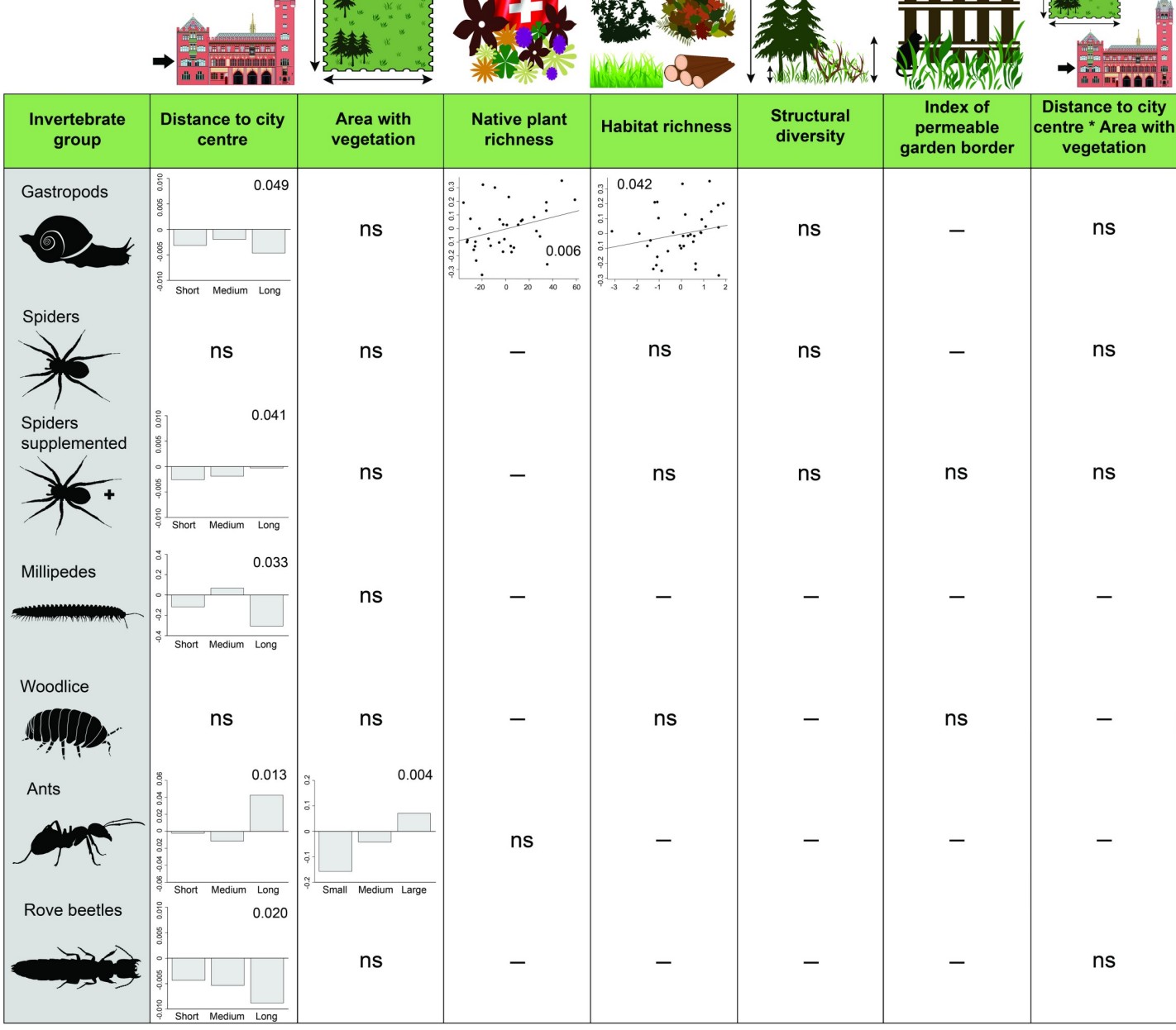

**Fig 1. Effects of urbanisation (distance to city centre; three classes), garden size (area with vegetation; three classes) and local garden characteristics on the species richness of six groups of invertebrates.** Plots show significant responses (P < 0.05) from GLMs (see Model 1 in S7 Table for more details). P-values for the response of the GLMs are shown. Displayed are deviance residuals for species richness from full models after stepwise reduction omitting the respective factor. This procedure corrected for other factors in the GLM. For native plant species richness, habitat richness and structural diversity, residuals from regressions of these factors on total garden area were used, because all three variables were correlated with garden size. Positive values in the bar plots indicate a higher than expected species richness. "–"indicates factors that were omitted from the models in the stepwise procedure. "ns" indicates factors that were retained in the model, but were not significant. For each family only represented by juvenile spiders, which were not identified to species level, we added an extra species to the count for the category spiders supplemented.

as classes (Fig 1; S7 Table), while structural diversity was positively related to gastropod richness in models with distance to city centre or percentage of sealed area as continuous variables (S7 Table).

## Effects of landscape and local garden characteristics on invertebrate abundance

Considering distance classes, distance to the city centre affected abundance of gastropods, spiders (incl. juveniles), woodlice (incl. juveniles) and rove beetles, but not that of millipedes (S7 Table). Considering distance to city centre as continuous variable, abundance of rove beetles was negatively influenced (S7 Table). Percentage of sealed area positively affected the abundance of rove beetles and spiders incl. juveniles (only models with classes), but not the abundance of the other groups (S7 Table). In models with percentage of sealed area as classes, rove beetle abundance was highest in gardens with a small area with vegetation and lowest in gardens with a medium-sized area with vegetation. In models with distance to the city centre as a continuous variable, abundance of adult spiders was positively influenced by area with vegetation (S7 Table). In contrast, millipede abundance was negatively affected by the area with vegetation in models with either distance to the city centre or percentage of sealed area as continuous variables (S7 Table). Increased border permeability had a positive effect on millipede abundance in the model with distance to city centre as classes, but did not influence other taxonomical groups (S7 Table). Surprisingly, millipede abundance was negatively affected by native plant species richness in all models (S7 Table). Increased habitat richness positively affected rove beetle abundance in the model with distance to the city centre as classes, and spider inclusively juveniles abundance in the three other models (S7 Table).

## Effects of landscape and local garden characteristics on invertebrate species composition

Principle coordinate analyses showed that gardens with different distances to the city centre (three classes) differed in species composition of millipedes, ants and rove beetles (Fig 2C, 2E and 2F) and tended to differ in gastropods (Fig 2A). Species composition of spiders and woodlice did not differ among gardens with different distance classes (Fig 2B and 2D). Similar results were obtained when percentage of sealed area in the surroundings (three classes) was used in the analysis instead of distance to city centre (S1 Fig).

PERMANOVAs revealed that structural diversity influenced the species composition of both gastropods ($F_{1,34}$ = 2.48, P = 0.022) and spiders ($F_{1,34}$ = 2.63, P = 0.008) in the gardens. The species composition of millipedes was affected by the area with vegetation ($F_{2,34}$ = 2.30, P = 0.041) and garden border permeability (index of permeable border: $F_{1,34}$ = 2.85, P = 0.040). Similarly, ant species composition was influenced by the area with vegetation ($F_{2,34}$ = 2.90, P = 0.035). In contrast, the species composition of woodlice and rove beetles were not affected by any local garden characteristics (in both cases P > 0.28). However, when singletons were excluded from the data set, then garden border permeability influenced species composition in rove beetles (index of permeability: $F_{1,34}$ = 3.04, P = 0.023).

The communities of the different invertebrate groups showed different distributions of similarity (contingency-test, 2136.6, d.f. = 45, P < 0.0001; S2 Fig). The average Sørensen similarity between two gardens ranged from 0.49 in ants, 0.54 in woodlice, 0.57 in gastropods, 0.59 in millipedes, 0.75 in spiders to 0.82 in rove beetles. Distance to the city centre (three classes) influenced the similarity in species composition in the invertebrate groups examined to a different extent (S3 Fig). The similarity in both the ant and rove beetle communities was lower in gardens in the centre of the city than in gardens at the periphery of the city (S3E and S3F Fig). In contrast, the similarity in gastropod communities was higher in gardens in the centre of the city than in gardens in the periphery of the city (S3A Fig). In spiders, millipedes and woodlice, the similarities of the communities were not affected by distance to city centre (S3B, S3 and S3D Fig). Similar results were obtained when percentage of sealed area in the surroundings

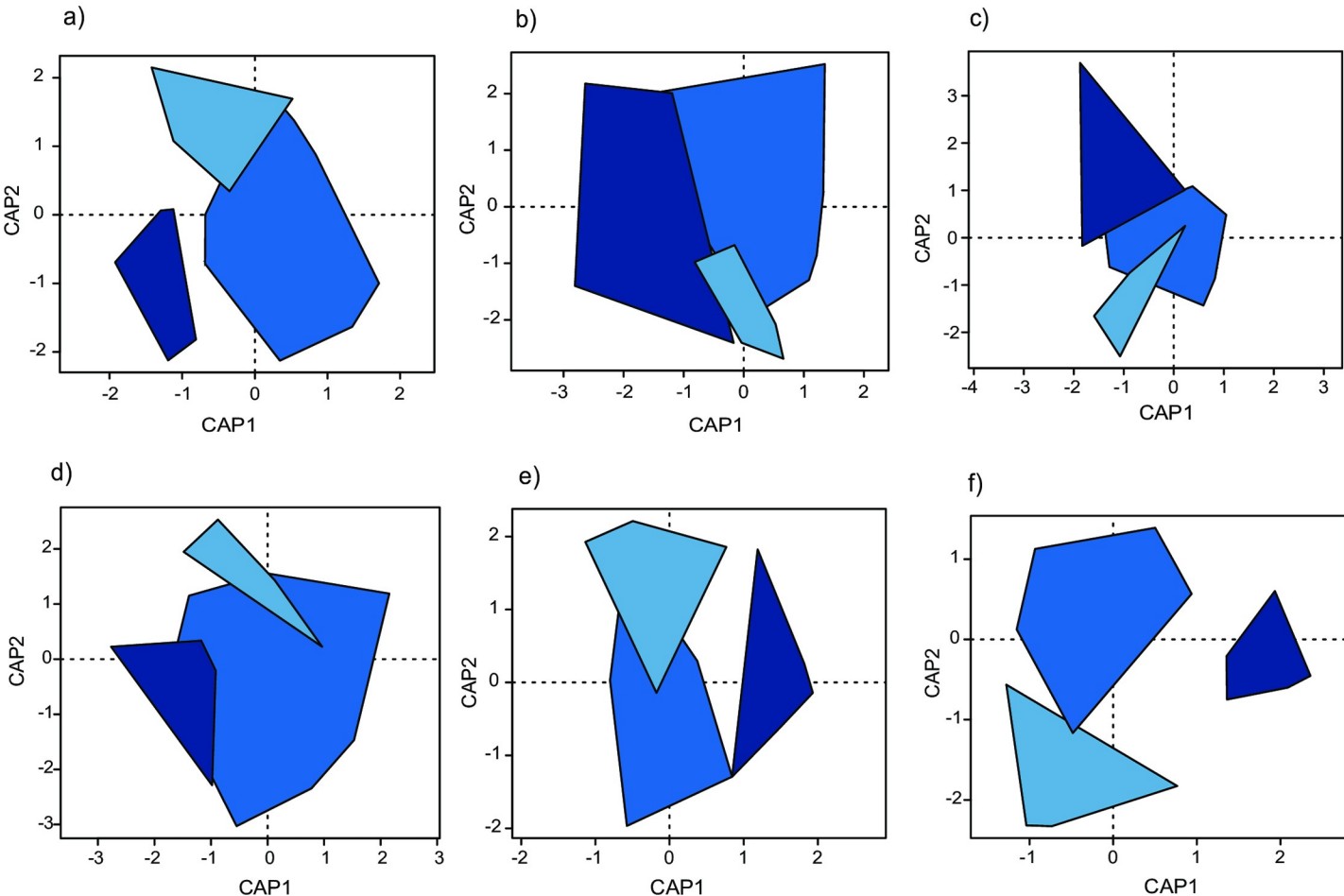

**Fig 2.** Results of constrained analyses of principle coordinates visualizing similarities in species compositions of gastropods (a), spiders (b), millipedes (c), woodlice (d), ants (e), and rove beetles (f) in gardens located at different distances to the city centre (three classes; dark blue refers to gardens in the centre of the city, blue to gardens at intermediate distance and light blue to gardens at long distance from the city centre).

(three classes) were used in the analysis instead of distance to city centre (S4 Fig), the only exception was that rove beetles were not affected by percentage of sealed area (S4F Fig).

## Discussion

### Biodiversity of urban gardens in Basel

Considered together, the 35 domestic gardens investigated in our study harboured a considerable share of the total Swiss species richness known for the corresponding groups (4.7–23.3%). This is impressive considering that the cumulated area of the 35 gardens amounted to 0.44 ha, just 0.000035% of the area of Switzerland (the area over which the gardens are spread represents only 0.2% of the area of Switzerland). Our study supports earlier findings that a highly variable mosaic of different habitat types, including an intensely managed one, can contain a significant part of a country's biodiversity [34]. Soil-dwelling invertebrates have been rarely included in studies of domestic gardens, an exception is Smith et al. [33], which reported similar shares of millipedes (17%), woodlice (24%) and ants (14%) in urban gardens of London as a percentage of the total species list for those groups across the whole of the British Isles. Island-biogeographic theory predicts that small islands harbour few species [51]. This is

partially based on the assumption that small islands have little habitat heterogeneity. In contrast, urban gardens are characterized by a small-grained mosaic of habitat types exceeding that of most similar-sized natural areas. Thus, domestic gardens provide a wide range of habitat niches within a small area, potentially providing essential resources for many species. This could explain the relatively high share of the species recorded in Switzerland that were found in the gardens examined.

The domestic gardens studied in Basel varied greatly in species richness. Unexpectedly, however, a particular garden harbouring a high proportion of the overall species richness recorded in one or a few taxonomical groups could have a poor proportion of the species richness recorded in other taxonomical groups. Thus, no garden was among the best suited for all invertebrate groups considered. This finding may be a result of a combination of several factors. Among them certain gardens may not fulfil the special requirements of the species of some taxonomical groups, e.g. because of intensive management or application of certain pesticides. For example, a lack of dead wood would preclude the presence of ant species constructing their nests in this habitat type [43, 80] and other saproxylic arthropods [89]. Furthermore, interactions among species of different taxonomical groups (some of those groups not examined in the present study) may affect the likelihood of coexistence. For example, competition for resources, predation as a controlling factor reducing abundance, and facilitation of some taxa through the presence of others (including species not examined) may all play a role. More precisely, intraguild interactions between spiders and ants with negative effects on ant abundances [90] and vice versa [91] are possible.

## Importance of landscape characteristics

Understanding how landscape characteristics affect biodiversity patterns and ecological processes at local and landscape scales is essential for promoting biodiversity in urban environments [3, 92]. We used the two landscape characteristics "distance to city centre" and "percentage of sealed area in the surroundings", which were related to different aspects of urbanisation. In Basel, distance to city centre is a rough estimate for the proximity to natural and semi-natural areas. In contrast, the percentage of sealed area mirrors the habitat matrix and thus inversely the percentage of green area in the proximity of a garden. In general, both landscape characteristics are inter-correlated, although significant deviations may occur as a result of decentralised secondary centres and industrialised areas. Both measures have been repeatedly used in studies of urban biodiversity patterns (e.g. [30, 93, 94]). Indeed, as in other studies, distance to city centre and percentage of sealed area were inter-correlated in Basel.

In our study, distance to the city centre of the gardens investigated ranged from 556 m to 9.5 km. The latter gardens were located in the rural surroundings, indicating that in an international context Basel is a small city (population of the city of Basel: 177,784 at the end of 2019; [95]; greater Basel area including Germany and France: 731,167; [96]). Considering percentage of sealed area, however, with a range of 32.8–87.0% in our study, this aspect of degree of urbanisation was comparable to that of much larger metropoles in Western Europe (Paris: 27.0–82.5% [97]; Sheffield, UK: 21–72% [34]; Zurich: 2.5–91.8% [98]). Interestingly, however, we found significant effects on both species richness and abundance more often in models with distance to the city centre than in those with percentage sealed area, even though the maximum distance to potential source populations in the surrounding rural areas was comparatively short. This suggests that the gradient in degree of urbanisation in Basel is relatively steep in relation to the distance from rural habitats. Factors associated with the distance to the city centre may act as filters decreasing the chance of establishment of certain species in gardens located in the centre of the city [99].

In our study, species richness and/or species composition of most groups were affected by distance to the city centre. However, the direction of the response varied among invertebrate groups. This supports findings from other studies, as reported in the review by Gosling et al. [100], in which 63.8% of studies on invertebrates showed a decrease in species richness with urbanisation and only 29.8% an increase (6.4% found no effect). In this context it is important to note that our approach differs from some other studies on the effects of urbanisation on species richness. We considered the same habitat type (domestic gardens) from the rural surroundings to the city centre. This contrasts with an alternative approach focusing on the urbanisation gradient by investigating plots occurring at given distances to the city centre. These plots may contain quite different habitat types (e.g. a nature preserve, recreational area, golf course, residential neighbourhood, office park and business district in Blair and Launer [101], and residential areas, golf courses and forest in Porter et al. [102]). Consequently, our gradient did not extent to natural or semi-natural habitats in rural areas, which may contain quite different species assemblages.

We found rather distinct species assemblages for gastropods, ants and rove beetles along the urbanisation gradient. Isolation of gardens in the centre of the city from habitats in the rural surroundings should reduce the probability of active colonisation, especially for less mobile species. Thus, the species assemblage in gardens in the centre of the city should primarily reflect local long-term conditions. While specimens of some of the taxonomical groups studied here actively or passively disperse through the air during at least a stage or part of their life cycle (flying rove beetles [103], ballooning juvenile spiders [104] and flying sexuals of ants [43, 80]), they are less mobile through most of their life cycle. This may explain the effect that distance to the city centre had on the species richness, abundance and species composition of these groups.

Species composition of both ants and rove beetles was changed in gardens in the centre of the city. Colonisation events in isolated gardens in the centre of the city are important in these groups as many rove beetles are able to fly and in ants males and young queens disperse by flying. This could explain the high variation in species composition among gardens in the centre of the city in those two taxonomical groups. In contrast, the less mobile gastropods showed high similarity in species composition among gardens in the centre of the city. A few generalist gastropod species (e.g. *Arion vulgaris* and *Hygromina cinctella*; [105]), some of them non-native, as well as millipede species [106] are frequently transported with ornamental plants and vegetables including soil around the roots. Once introduced, these and other disturbance-tolerant species may persist in these habitats.

## Importance of garden size and other local garden characteristics

Size of private domestic gardens depends on historical city development, cultural aspects and traditions, economic factors and owner preferences [28, 107], and thus may vary among cities [21]. The gardens examined in our study (mean area: 480 m$^2$) were larger than domestic gardens investigated in Sheffield (173 m$^2$; [108]), five cities in the UK (289 m$^2$; [109]), and in the Greater Toronto Area, Canada (311 m$^2$; [110]), but smaller than community gardens in California [111] and New York (Harlem and Bronx) [30]. However, in our study, gardens in the centre of the city (class "short distance to city centre": 163 m$^2$) were comparable in size to those of Sheffield (163 m$^2$). Gardens in the suburban belt and the surroundings of Basel on average were larger (class "medium distance to city centre": 533 m$^2$, class "long distance to city centre": 527 m$^2$) even though the size range included gardens comparable to gardens in the centre of the city.

Garden size is of importance as suggested by the general relationship between species richness and area (e.g. [112] for plants). Furthermore, management and planting decisions may depend on garden size [107]. Thus, various garden characteristics may also be influenced by garden size, indirectly affecting the local biodiversity. Indeed, we found that almost all garden

characteristics considered, including habitat richness and structural diversity, were correlated with garden size; an exception being index of permeable border. This may confound analyses of species richness. We circumvented the problem by calculation of regressions of garden characteristics on garden size and using the residuals of these relationships for our models. Furthermore, in models analysing the effects of distance to city centre or percentage of sealed area in the surroundings on species richness and abundance, we always considered area with vegetation as a measure of habitat size.

Compared to the landscape characteristic "distance to the city centre", local garden characteristics had less power in explaining patterns of species richness and abundance in the different invertebrate groups studied. Yet, in our study, species richness of spiders and ants and abundance of spiders, millipedes and rove beetles were related to garden size (represented by area with vegetation), while native plant species richness influenced gastropod and ant species richness and millipede abundance. The effect of garden size on the diversity of ground-dwelling invertebrates has rarely been studied (but see e.g. [27, 34, 99, 113]). Negative effects of garden size on beetle species richness [113] and positive effects on harvestmen abundance [34] were observed in domestic gardens in Sheffield.

Interestingly, neither species richness nor abundance of the groups examined were related to the index of permeable border, an exception being millipede abundance. This indicates that the borders of the gardens examined did not function as absolute barrier for most of the invertebrate groups examined. If neither index of permeable border nor garden size have an effect on the diversity of an invertebrate group, then this may indicate that the gardens should be considered as a functional unit with adjacent gardens as suggested by Smith et al. [113] and Goddard et al. [15]. In this context it is important to consider that even though most of the taxonomical groups studied had low dispersal ability during most of their life cycle. As explained above, some of them have a short period of aerial dispersal (ballooning spiders, ant mating flights), while some rove beetle species disperse actively [103], and snails may occasionally be dispersed passively attached to leave litter transported by wind [37]. Furthermore, species may be brought into gardens passively as garden owners import plants or soil [105, 106]. In such cases, the nature of the borders would not be relevant.

Contrary to our hypothesis, habitat richness and structural diversity only affected gastropod species richness. Structural diversity of domestic gardens has been found to influence arachnid richness by Smith et al. [113] and even more mobile groups such as bumblebees [114] and birds [115] (not examined in this study). Populations of ground-dwelling invertebrates may respond less quickly to year-by-year variation in garden structural diversity than those of flying species. Various habitat features have been shown to positively influence the diversity of certain taxonomic groups in gardens [113]. This indicates that a rich variety of different habitat types should lead to high overall species richness. In our study, however, only gastropods species richness was influenced by habitat richness across the groups examined.

## Advantage of using multiple taxa in a biodiversity survey

The diversity of a particular taxonomic group may not mirror the overall biodiversity [116]. Different taxonomical groups may respond to the same factor to a different degree or at different spatial scales. Furthermore, different taxa have different resource needs and habitat requirements. Yet, estimates of biodiversity are frequently based on the species richness of one or a few easily studied indicator groups (e.g. vascular plants, butterflies or birds), which may even partly depend on each other [116, 117]. The power of such an approach depends on the indicators chosen to match the scale of the investigation unit. For mobile species such as wild bees, butterflies, and birds with large home ranges a single garden may only constitute a part of

their home range or territory. We therefore focused on taxonomical groups, which are characterised by species of relatively low mobility, and which are able to complete their entire life cycle within a garden, thus matching the scale of an average domestic garden. The invertebrate groups examined in our study have no close specific relationships among each other (exceptions may occur at species level). Furthermore, these inconspicuous small invertebrates are often not noticed by the garden owners and not actively promoted or intentionally transported by them.

It is frequently assumed that the diversity of an indicator group is correlated with the diversity of other groups. Studies that tested intercorrelations among groups reported positive associations between species richness of vascular plants and butterflies, two groups with often specific interactions. In contrast, such associations were not found among many other taxonomical groups [116, 118, 119]. No correlation in species richness is expected in groups with a high proportion of generalist species or whose species do not have close specific interactions with species from other groups [119]. These two characteristics describe the groups examined in our study, which to date have rarely been studied in combination. Indeed, our findings show that species richness per garden was typically not correlated among the different invertebrate groups we examined. An exception was ants, whose richness was correlated with that of gastropods and spiders. One could expect ants and spider richness to respond positively to a high structural diversity in the gardens, however, our data do not provide evidence for this. Our findings suggest that in an ideal case, a biodiversity assessment is not based on a single indicator group but on several taxonomical groups with a range of different habitat requirements and belonging to different trophic levels [118, 119].

## Conclusions and outlook

We considered explicitly invertebrate groups that are not promoted by garden owners. Previous studies on biodiversity in domestic gardens usually focused on other groups, such as flowering plants, wild bees, butterflies and birds, whose diversity may at least partly reflect larger scale habitat diversity. The groups considered in our study are characterised by a low mobility of most of the constituent species, which is more in line with the spatial scale of domestic gardens than the home range of species from more mobile groups such as flying insects and birds. Nonetheless, we recorded relatively few effects of local garden characteristics on the richness and abundance in most of the groups examined. An exception was gastropods, which might be the group with the lowest active mobility. Indeed, most invertebrate groups were rather affected by landscape characteristics, in particular by distance to the city centre, suggesting that factors associated with this variable act as filter for the establishment of certain species.

Our study indicates that single domestic gardens, as part of a network of green infrastructure, might be of importance for the maintenance of regional biodiversity. Complementing our study with findings from other work on more mobile taxa in urban gardens, we suggest that garden owners can improve conditions for many species by increasing habitat diversity and implementing biodiversity-friendly management practices [120] and, for example, by replacing exotic plants with native species [31]. At a larger spatial scale, urban planners should consider the valuable contributions made by the mosaic of highly variable domestic gardens and need to promote the biodiversity therein by reducing urbanisation in the surroundings through measures like urban greenways increasing biological connectivity [12, 13, 20].

## Supporting information

**S1 Table. Species richness correlations among taxonomical groups within 35 gardens (Pearson correlations (R) or Spearman correlations ($R_s$)).** P-values are given in brackets. (DOCX)

**S2 Table. Means and ranges for garden and landscape characteristics (n = 35 in all cases).**
(DOCX)

**S3 Table. Local garden characteristics and landscape characteristics per garden.**
(XLS)

**S4 Table. Abundance of invertebrate species recorded in 35 domestic gardens.** For ants presence data is given.
(XLSX)

**S5 Table. Species richness and percentage of the species pool of all gardens for seven taxonomical groups of ground-dwelling invertebrates for 35 domestic gardens.**
(XLS)

**S6 Table. Numbers of specimens collected for each of six taxonomical groups in 35 domestic gardens.**
(XLS)

**S7 Table. Summaries of GLMs testing the effects of landscape factors (distance to city centre or percentage of sealed area in the surroundings), garden size (area with vegetation) and local garden characteristics, as well as of the interaction between the landscape factors and garden size on species richness and abundance of different taxonomical groups.** As the two landscape factors were intercorrelated, separate models were used to assess their effects. To test whether the assignment to classes with different distance to city centre, different percentages of sealed area in the surroundings, or different garden sizes affected the outcomes, analyses were repeated using models that treated all factors as continuous variables. This resulted in four different model types (models 1–4). Full models were stepwise reduced by omitting variables explaining little variation ($F < 1.0$) staring with the variable with the lowest F-value. However, the main landscape factors and garden size were always retained in the model. Species richness was log-transformed. All models used a quasipoission error distribution and log-link function.
(XLSX)

**S1 Fig.** Results of constrained analyses of principle coordinates visualizing similarities in species compositions of gastropods (a), spiders (b), millipedes (c), woodlice (d), ants (e), and rove beetles (f) in gardens with different percentages of sealed area within a radius of 200 m (three classes). Dark red refers to gardens with a high percentage of sealed area, red to gardens with intermediate percentage of sealed area and light red to gardens with low percentage of sealed area in the surroundings.
(PDF)

**S2 Fig.** Distribution of Sørensen-indices of species compositions obtained from all combinations of each two gardens for gastropods (a), spiders (b), millipedes (c), woodlice (d), ants (e), and rove beetles (f).
(PDF)

**S3 Fig.** Effects of distance to the city centre (three classes) on the Sørensen-indices of species compositions of gastropods (a), spiders (b), millipedes (c), woodlice (d), ants (e), and rove beetles (f). The Sørensen-indices of species compositions were calculated for all combinations of each two gardens belonging to the same distance class. Different letters indicate significant differences among distance classes (Tukey's HSD, $P < 0.05$).
(PDF)

**S4 Fig.** Effects of percentage of sealed area within a radius of 200 m (three classes) on the Sørensen-indices of species compositions of gastropods (a), spiders (b), millipedes (c), woodlice (d), ants (e), and rove beetles (f). The Sørensen-indices of species compositions were calculated for all combinations of each two gardens belonging to the same distance class. Different letters indicate significant differences among distance classes (Tukey's HSD, $P < 0.05$).
(PDF)

## Acknowledgments

We in particular thank the owners for allowing us to access their gardens for this study. We thank B. Feldmann for rove beetle identifications, M. Raupach for woodlice identifications, K Hannig for ground beetle identifications, V. Ingold and C. Ramage for assistance with the fieldwork, and V. Martínez-Pillado for her help with the preparation of Fig 1. We thank three anonymous reviewers for comments on an earlier version of this manuscript.

## Author Contributions

**Conceptualization:** Brigitte Braschler, José D. Gilgado, Hans-Peter Rusterholz, Bruno Baur.

**Data curation:** Brigitte Braschler.

**Formal analysis:** Brigitte Braschler, Hans-Peter Rusterholz.

**Funding acquisition:** José D. Gilgado, Bruno Baur.

**Investigation:** Brigitte Braschler, José D. Gilgado, Valerie Zwahlen, Hans-Peter Rusterholz, Bruno Baur.

**Methodology:** Brigitte Braschler, José D. Gilgado, Valerie Zwahlen, Hans-Peter Rusterholz, Sascha Buchholz, Bruno Baur.

**Supervision:** Hans-Peter Rusterholz, Bruno Baur.

**Visualization:** Brigitte Braschler, José D. Gilgado, Hans-Peter Rusterholz.

**Writing – original draft:** Brigitte Braschler, Hans-Peter Rusterholz, Bruno Baur.

**Writing – review & editing:** Brigitte Braschler, José D. Gilgado, Valerie Zwahlen, Hans-Peter Rusterholz, Sascha Buchholz, Bruno Baur.

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
