## [Decision Letter · Decision Letter 0]

23 Jul 2020

PONE-D-20-14585

Ground-dwelling invertebrate diversity in domestic gardens along a rural-urban gradient: landscape characteristics are more important than garden characteristics

PLOS ONE

Dear Dr. Braschler,

Thank you for submitting your manuscript to PLOS ONE. After careful consideration, we feel that it has merit but does not fully meet PLOS ONE’s publication criteria as it currently stands. Therefore, we invite you to submit a revised version of the manuscript that addresses the points raised during the review process.

The study is interesting. Minor revision and reasonable response to the comments of the reviewers are needed.

We look forward to receiving your revised manuscript.

Kind regards,

Jian Liu

Academic Editor

PLOS ONE

Journal Requirements:

Reviewers' comments:

Reviewer's Responses to Questions

**Comments to the Author**

1. Is the manuscript technically sound, and do the data support the conclusions?

Reviewer #1: Yes

Reviewer #2: Yes

Reviewer #3: Yes

2. Has the statistical analysis been performed appropriately and rigorously? 

Reviewer #1: Yes

Reviewer #2: Yes

Reviewer #3: Yes

3. Have the authors made all data underlying the findings in their manuscript fully available?

Reviewer #1: Yes

Reviewer #2: Yes

Reviewer #3: Yes

4. Is the manuscript presented in an intelligible fashion and written in standard English?

Reviewer #1: Yes

Reviewer #2: Yes

Reviewer #3: Yes

5. Review Comments to the Author

Reviewer #1: Please see my comments on the manuscript. They are minor and relate primarily to a few typos and some clarification with the methods/study design. This is a very thorough study and increasingly important as urban development continues in concert with new planning/conservation strategies to mitigate its impact on biodiversity through the development of more connected and well managed urban green networks. The results were surprising especially as they relate to how gardens capture a high percentage of the biodiversity present in the country despite the relatively small garden areas. Overall I think the manuscript is well written, the results are well defines (I especially appreciate the table and figures), and theresults add to the growing body of literature on urban conservation - particularly when studies focusing on invertebrates often do not include these important functional groups.

Reviewer #2: The paper is exceptionally well written and the study brings important information into the growing field of urban ecology. The study would have been greatly improved by giving more consideration to human factors: historical development patterns, socioeconomic status, cultural diversity, resident knowledge of gardening, and resident self-report of maintenance. The use of an urban gradient created a useful comparative sample for a deeper understanding of human interventions into the success of microecosystem fragments across the city.

Reviewer #3: Ground-dwelling invertebrate diversity in domestic gardens along a rural-urban gradient: landscape characteristics are more important than garden characteristics

Braschler et al.

Braschler et al. used a multi-taxa approach to examine patterns of invertebrate biodiversity across the urban to rural gradient. Overall, I think there methodology was sound. The results allow landscape ecologists and designers to think about the relative effects of different landscape and garden factors on invertebrate biodiversity.

While I think the study is sound and worthy of publication. I found that rewriting the Discussion would be helpful. Much of the discussion either reiterated the Introduction or Results, without providing much additional explanations as to why these ecological patterns may occur. For instance, the authors found that the landscape and garden affected invertebrate taxa differently. However, they do not provide much explanation as to why this may occur. I think the paper would be richer and more interesting if they discussed the evidence on how mobility may differ by taxa.

Introduction

Line 99. A reference is needed here.

Line 101. A reference is also needed here.

Line 126. The term “naturalness” is not well defined. A more quantifiable term should be used. It is unclear if the authors are referring to the management style, proportion of indigenous species, level of inputs, etc.

Line 130. This sentence sounds more like a prediction rather than a hypothesis.

Line 134. This paragraph departs from the ongoing narrative and should be linked better with the rest of the Introduction.

Methods

I am wondering how representative the gardens are. Could the volunteers be somehow biased towards people who value nature, and thus, the garden management and species composition would be reflective of this bias?

I am also wondering how some of the metrics in Table 1 were scaled by area. The total native plant species per area would be particularly susceptible to the overall size of the gardens.

Line 175. Change ‘is’ to ‘was’.

Line 201. Change to ‘tape measure’.

Line 382-383. Explain the logic further on the approach of using the “residuals of the relationships between the variable and total garden area”.

Results

Lines 458-465. I found that there has been little discussion as to how many species “should be found in gardens”. The results presented showed a range of 4.7-23% of the total number of species found in Switzerland. For some of these groups that are more diverse, such as ground beetles, the % of total metric may actually be a better indicator of sensitivity to disturbance. The less speciose taxonomic groups may be less sensitive if there are more generalists.

Line 467-469. I found myself looking at the “mean percentage” category. To figure out how the mean percentages could end up totaling above 100%. Maybe I missed something about how this was calculated. Does a mean percentage of 40.6 ±17.1 mean that among all of the individuals specimens collected, 40.6% of the individuals were woodlice?

Lines 512-515. I didn’t understand what the phrase meant “most landscape effects disappeared if the percentage of sealed area…is used in the models”. Please rephrase this sentence more clearly.

Line 589. I suggest that the authors rethink the naming of the city centers as something other than the “inner city”, because that is a racially-tinged term. The use of the term “inner city” is negative in America. In America, the term is used to refer to the “the usually older, poorer, and more densely populated central section of a city”. (Merriam-Webster). The US government had years of anti-black racist policies associated with housing that caused the city centers to be mostly poor people non-white people. I also noticed how in Fig. 2 the urban gardens are black and the shades lighten with distance out of the city center.

Discussion

Overall, I found that the Discussion was a bit repetitive of the Introduction. In many cases some of the parts of the Discussion repeated the Introduction and Results sections instead of focusing explaining the results.

I was hoping to hear a broader perspective on invertebrate biodiversity. I suggest that the discussion to be reorganized to clearly explain the major factors that influence garden biodiversity. In the end, I found myself confused by the relative effects of the matrix (proximity to natural or semi-natural habitats) or local garden effects.

It seems like it would be useful to review more work from Teja Tscharntke, Doug Landis, and other landscape ecologists to compare this idea of local management effects vs. landscape effects on biodiversity.

Line 618-620. I find myself wondering what is the expected of biodiversity on small parcels of land. Are there any similar surveys that have tried to generate species accumulation/area curves? Without these estimates it seems quite subjective to say that this is a considerable share of the total species richness.

Line 634. Add the phrase “combined effect”. While I appreciate the possible factors, they remain speculative due to the lack of concrete evidence. I think it would be useful to have references here.

Line 646. Add “the percentage of sealed area”.

Line 732-733. It would be helpful to discuss further how increases in native plant species had variable effects on ant (higher), gastropod (higher), and millipede diversity (lower). In the discussion, I found that there was little general discussion on why.

Line 775-779. These lines repeated part of the Introduction and should be moved there. As a reader, I am hoping to have the authors explain why the diversity of the indicator groups are not correlated. What mechanisms may be operating?

6. PLOS authors have the option to publish the peer review history of their article (what does this mean?). If published, this will include your full peer review and any attached files.

Reviewer #1: No

Reviewer #2: No

Reviewer #3: No

---

## [Author Response · Author response to Decision Letter 0]

27 Aug 2020

27 August 2020

PONE-D-20-14585

Revision of " Ground-dwelling invertebrate diversity in domestic gardens along a rural-urban gradient: Landscape characteristics are more important than garden characteristics". 

Following the requests by the journal we made the following changes. We now provide two new figures in the supplementary material, showing the results of the rda and the Sörensen similarity analyses using percentage of sealed area rather than distance to the city centre to delineate classes, replacing the statement “data not shown”. We also reformatted the titles, subtitle, figure refererences and captions and the corresponding author reference according to the templates. The Section of Conservation Biology is indeed a subunit of the Department of Environmental Sciences and thus the affiliation is correct in order from small to large as stated.

No permit from a government institution was necessary to conduct the field work for this study on invertebrates within settlements. Access to the study sites (gardens) was granted by the respective owners offering access to the garden of their private domiciles. For data protection we do not list these owners by name in the paper, and do not include precise coordinates or maps for the sites in the manuscript. We now clarify this in the methods section of the manuscript (lines 178–186).

We considered the remarks of the reviewers and made the following improvements.

Reviewer #1: Please see my comments on the manuscript. They are minor and relate primarily to a few typos and some clarification with the methods/study design. This is a very thorough study and increasingly important as urban development continues in concert with new planning/conservation strategies to mitigate its impact on biodiversity through the development of more connected and well managed urban green networks. The results were surprising especially as they relate to how gardens capture a high percentage of the biodiversity present in the country despite the relatively small garden areas. Overall I think the manuscript is well written, the results are well defines (I especially appreciate the table and figures), and theresults add to the growing body of literature on urban conservation - particularly when studies focusing on invertebrates often do not include these important functional groups.

Answer: We considered the remarks of the reviewer inserted in the pdf-document and improved the manuscript. Two points need a special reply.

- Comment on line 169. We did not measure the distance to the nearest other green space because this measure was in most cases 0 metres (adjoining neighbour garden).

- Table 1. We did not assess species richness of horticultural plants because many plants were represented by several varieties in single gardens. Varieties are representing another taxonomical rank below species.

Reviewer #2: The paper is exceptionally well written and the study brings important information into the growing field of urban ecology. The study would have been greatly improved by giving more consideration to human factors: historical development patterns, socioeconomic status, cultural diversity, resident knowledge of gardening, and resident self-report of maintenance. The use of an urban gradient created a useful comparative sample for a deeper understanding of human interventions into the success of microecosystem fragments across the city.

A: We did not consider human factors for the following reasons. Firstly, historical development patterns: in our sample of 35 gardens, many of the owners have been responsible for the gardens only for a few years and no information was available on previous management activities that may still affect local diversity (e.g. former application of pesticides). Secondly, socioeconomic status of home owners does not represent the entire range. Switzerland has a very high percentage of tenants compared to other countries with home ownership mostly reserved for people of higher economic status. The same argument applies to cultural diversity. We agree with the reviewer that data on the residents’ maintenance of their property would be of interest. However, we noted that residents’ self report of maintenance was unreliable in several cases. Garden owners varied widely in what they considered as poison, and showed difficulties providing details on mowing frequency. We did not analyse this information because of these uncertainties. In a follow-up project one year later, we used structured interviews to assess garden owners’ knowledge and views of biodiversity and gardening. Analysis of this data is in progress and will be reported elsewhere.

Reviewer #3: Ground-dwelling invertebrate diversity in domestic gardens along a rural-urban gradient: landscape characteristics are more important than garden characteristics

Braschler et al. used a multi-taxa approach to examine patterns of invertebrate biodiversity across the urban to rural gradient. Overall, I think there methodology was sound. The results allow landscape ecologists and designers to think about the relative effects of different landscape and garden factors on invertebrate biodiversity.

While I think the study is sound and worthy of publication. I found that rewriting the Discussion would be helpful. Much of the discussion either reiterated the Introduction or Results, without providing much additional explanations as to why these ecological patterns may occur. For instance, the authors found that the landscape and garden affected invertebrate taxa differently. However, they do not provide much explanation as to why this may occur. I think the paper would be richer and more interesting if they discussed the evidence on how mobility may differ by taxa.

A: We reworked the discussion section to add more context on factors potentially causing the patterns observed in this study (see below).

Introduction

Line 99. A reference is needed here.

A: We inserted an adequate reference ([30] Matteson KC, Langelotto GA. Determinates of inner city butterfly and bee species richness. Urban Ecosyst. 2010; 13:333–347.

Line 101. A reference is also needed here.

A: We inserted three references [Baur 1989, Baur & Baur 1988, David 2009]. 

Line 126. The term “naturalness” is not well defined. A more quantifiable term should be used. It is unclear if the authors are referring to the management style, proportion of indigenous species, level of inputs, etc.

A: We rephrased the sentence to reflect the fact that at this point we still refer to a composite impression of the garden. We therefore added “proportion of native plant species, presence of wildlife friendly features such as dead wood or stone piles, extensive management of grassland, bushes and hedges”.

Line 130. This sentence sounds more like a prediction rather than a hypothesis.

A: We rephrased this sentence.

Line 134. This paragraph departs from the ongoing narrative and should be linked better with the rest of the Introduction.

A: We improved this paragraph by adding two introductory sentences.

Methods

I am wondering how representative the gardens are. Could the volunteers be somehow biased towards people who value nature, and thus, the garden management and species composition would be reflective of this bias?

A: As stated in the methods, gardens were selected from a pool 65 candidates offered in response to public calls at a local conference, in a municipal newspaper and a newsletter, as well as through personal contacts of the authors. Out of these 65 gardens we chose 35 gardens that reflected a rural-urban gradient and represented both a range of garden sizes and different management (little to intensively managed) spread along the urbanisation gradient. Further criteria for the garden choice were acceptance of the intended sampling methods by the garden owners and guaranteed daytime access to the gardens. All garden owners were interested in their gardens but for different purposes (recreation, cultivation of flowers, fruit and vegetables, landscaping or biodiversity).

I am also wondering how some of the metrics in Table 1 were scaled by area. The total native plant species per area would be particularly susceptible to the overall size of the gardens.

A: Indeed many variables were correlated with garden size. As explained in the manuscript, we thus used the residuals of relationsships between those variables and the total garden area for analysis rather than the raw variable. This is a standard procedure to correct for size relationships. In this way, the factor in the model was thus statistically corrected for garden size, so we could analyse its effect independent of garden size.

Line 175. Change ‘is’ to ‘was’.

A: Done

Line 201. Change to ‘tape measure’.

A: Done

Line 382-383. Explain the logic further on the approach of using the “residuals of the relationships between the variable and total garden area”.

A: Many variables were correlated with garden size. As explained in the manuscript, we thus used the residuals of relationsships between those variables and the total garden area rather than the raw variable. This is a standard procedure to correct for size relationships. In this way, the factor in the model was thus statistically corrected for garden size, so we could analyse its effect independent of garden size.

Results

Lines 458-465. I found that there has been little discussion as to how many species “should be found in gardens”. The results presented showed a range of 4.7-23% of the total number of species found in Switzerland. For some of these groups that are more diverse, such as ground beetles, the % of total metric may actually be a better indicator of sensitivity to disturbance. The less speciose taxonomic groups may be less sensitive if there are more generalists.

A: We do not understand the phrase “should be found in gardens”. We considered the second part of this comment in our revision of the discussion.

Line 467-469. I found myself looking at the “mean percentage” category. To figure out how the mean percentages could end up totaling above 100%. Maybe I missed something about how this was calculated. Does a mean percentage of 40.6 ±17.1 mean that among all of the individuals specimens collected, 40.6% of the individuals were woodlice?

A: No the values in the column are not meant to be added up. Rather the percentages refer to the share of overall biodiversity within a taxonomic group that was represented in a single garden. The value cited by the reviewer thus indicates that on average 40.6% of woodlice species found overall in the project (all gardens combined) were present in a single garden, though as the SD indicates there was some variation among gardens with some gardens having a smaller or larger share of the overall woodlice species richness represented. To clarify this point we improved the caption of Table 2.

Lines 512-515. I didn’t understand what the phrase meant “most landscape effects disappeared if the percentage of sealed area…is used in the models”. Please rephrase this sentence more clearly.

A: We clarified this point by slightly rephrasing the sentence. In particular we state that the landscape effects of the models mentioned above disappeared when percentage of sealed area was included as a proxy for urbansitation rather than distance to the city centre.

Line 589. I suggest that the authors rethink the naming of the city centers as something other than the “inner city”, because that is a racially-tinged term. The use of the term “inner city” is negative in America. In America, the term is used to refer to the “the usually older, poorer, and more densely populated central section of a city”. (Merriam-Webster). The US government had years of anti-black racist policies associated with housing that caused the city centers to be mostly poor people non-white people. I also noticed how in Fig. 2 the urban gardens are black and the shades lighten with distance out of the city center.

A: We replaced “inner city” with “in the centre of the city” throughout. Furthermore, we replaced the grey scale in Figure 2 and in the new supplementary figure S1 Figure by shades of blue and red respectively.

Discussion

Overall, I found that the Discussion was a bit repetitive of the Introduction. In many cases some of the parts of the Discussion repeated the Introduction and Results sections instead of focusing explaining the results.

A: We improved the discussion section following the advise of the reviewer (see our responses to the comments below).

I was hoping to hear a broader perspective on invertebrate biodiversity. I suggest that the discussion to be reorganized to clearly explain the major factors that influence garden biodiversity. In the end, I found myself confused by the relative effects of the matrix (proximity to natural or semi-natural habitats) or local garden effects.

A: We inserted details on invertebrate biodiversity (see also below). Furthermore, we added information on the relative effects of the matrix.

It seems like it would be useful to review more work from Teja Tscharntke, Doug Landis, and other landscape ecologists to compare this idea of local management effects vs. landscape effects on biodiversity.

A: Following the advise of the reviewer we inserted a key reference by these authors on the importance of considering the effects of landscape composition on local characteristics.

Line 618-620. I find myself wondering what is the expected of biodiversity on small parcels of land. Are there any similar surveys that have tried to generate species accumulation/area curves? Without these estimates it seems quite subjective to say that this is a considerable share of the total species richness.

A: Species-area accumulation curves are not the focus of this study. We improved the manuscript instead by explaining the importance of habitat mosaics for increasing biodiversity in small areas.

Line 634. Add the phrase “combined effect”. While I appreciate the possible factors, they remain speculative due to the lack of concrete evidence. I think it would be useful to have references here.

A: We rephrased this paragraph and inserted several examples with corresponding references.

Line 646. Add “the percentage of sealed area”.

A: Done

Line 732-733. It would be helpful to discuss further how increases in native plant species had variable effects on ant (higher), gastropod (higher), and millipede diversity (lower). In the discussion, I found that there was little general discussion on why.

A: This is a misinterpretation by the reviewer. We found no contrasting effects on species richness in different groups. Millipede abundance (not millipede diversity as the reviewer states) was negatively affected by native plant species richness.

Line 775-779. These lines repeated part of the Introduction and should be moved there. As a reader, I am hoping to have the authors explain why the diversity of the indicator groups are not correlated. What mechanisms may be operating?

A: We rephrased the paragraph. The paragraph is thus no longer a repeated part of the introduction. We explained under which conditions association between species richness of two taxonomical groups can be expected. We show that these conditions were not fullfilled by our focal groups. We also present a possible explanation for the association between ants and spiders. 

We thank the reviewers for their comments and Dr Jian Liu and the PLoS ONE team for the editorial work. Hereby we resubmit the revised manuscript.

Kind regards,

Brigitte Braschler

---

## [Decision Letter · Decision Letter 1]

18 Sep 2020

Ground-dwelling invertebrate diversity in domestic gardens along a rural-urban gradient: Landscape characteristics are more important than garden characteristics

PONE-D-20-14585R1

Dear Dr. Braschler,

We’re pleased to inform you that your manuscript has been judged scientifically suitable for publication and will be formally accepted for publication once it meets all outstanding technical requirements.

Kind regards,

Jian Liu

Academic Editor

PLOS ONE

Additional Editor Comments (optional):

Reviewers' comments:

Reviewer's Responses to Questions

**Comments to the Author**

1. If the authors have adequately addressed your comments raised in a previous round of review and you feel that this manuscript is now acceptable for publication, you may indicate that here to bypass the “Comments to the Author” section, enter your conflict of interest statement in the “Confidential to Editor” section, and submit your "Accept" recommendation.

Reviewer #1: All comments have been addressed

Reviewer #2: All comments have been addressed

2. Is the manuscript technically sound, and do the data support the conclusions?

Reviewer #1: Yes

Reviewer #2: Yes

3. Has the statistical analysis been performed appropriately and rigorously? 

Reviewer #1: Yes

Reviewer #2: Yes

4. Have the authors made all data underlying the findings in their manuscript fully available?

Reviewer #1: Yes

Reviewer #2: Yes

5. Is the manuscript presented in an intelligible fashion and written in standard English?

Reviewer #1: Yes

Reviewer #2: Yes

6. Review Comments to the Author

Reviewer #1: I am satisfied with the overall corrections made to the manuscript. The majority of the suggested revisions and clarifications were minor.

Reviewer #2: None. Given that this is an "urban" paper, i.e. people, I'm looking forward to seeing the human-factors published, granted, elsewhere.

7. PLOS authors have the option to publish the peer review history of their article (what does this mean?). If published, this will include your full peer review and any attached files.

Reviewer #1: **Yes: **Jaret C. Daniels

Reviewer #2: No

---

## [Editor Report · Acceptance letter]

25 Sep 2020

PONE-D-20-14585R1 

Ground-dwelling invertebrate diversity in domestic gardens along a rural-urban gradient: Landscape characteristics are more important than garden characteristics 

Dear Dr. Braschler:

I'm pleased to inform you that your manuscript has been deemed suitable for publication in PLOS ONE. Congratulations! Your manuscript is now with our production department. 

Kind regards, 

on behalf of

Dr. Jian Liu 

Academic Editor

PLOS ONE